# Selenium Biofortification: Strategies, Progress and Challenges

Ofori Prince Danso [1,2], Bismark Asante-Badu [3], Zezhou Zhang [4], Jiaping Song [4], Zhangmin Wang [2], Xuebin Yin [1,2,4,5,*] and Renbin Zhu [1]

1   School of Earth and Space Sciences, Anhui Province Key Laboratory of Polar Environment and Global Change, University of Science and Technology of China, Hefei 230026, China
2   Nanjing Institute for FAST (iFAST), National Innovation Center for Functional Rice, Nanjing 210031, China
3   Institute of Agricultural Resources and Regional Planning, Graduate School of Chinese Academy of Agricultural Sciences (GSCAAS), Beijing 100081, China
4   College of Resource and Environment, Anhui Science and Technology University, 9 Donghua Road, Fengyang County, Chuzhou 233100, China
5   Institute for FAST (iFAST) at Yangtze River Delta, Chuzhou 239000, China
*   Correspondence: xbyin@ahstu.edu.cn

**Abstract:** Selenium (Se) is an essential trace element for humans and animals. Its necessity for plants is still under examination. Due to the contradictory nature of Se and its significance, it has received much interest in recent years. Se deficiency can be harmful to humans, yet almost a billion people are deficient. Its deficiency has been associated with cancers, impairment of organs, and a number of other ailments. The biofortification of plants and livestock is a guaranteed practice to increase human selenium consumption. Strategies such as foliar spraying, the direct application of Se in plants and Se feed, and injections in livestock have been employed. Se biofortification has been shown to have additional beneficial effects in plants and livestock. In plants, it has been reported to mitigate different types of stress and increase yield. In animal biofortification, Se has been shown to reduce the detrimental effects of ailments and promote healthy growth. Se biofortification, nevertheless, confronts a number of difficulties. For instance, the bulk of biofortified products must be prepared before consumption, lowering the Se concentration. The objective of this review is to convey the current understanding of the Se biofortification of plants and animals, as well as its difficulties, taking into account both the detrimental consequences of Se deficiency and benefits of Se biofortification.

**Keywords:** selenium; biofortification; plants; livestock; food; humans

## 1. Introduction

Selenium (Se) is mainly generated as a byproduct of copper mining [1]. Since its industrial use began in the early 1900s, the global output of Se has expanded significantly. Worldwide production in 1910 was around 5000 kg [1]. According to Garside, about 3300 metric tons of Se was produced globally in 2020. China, Japan, and Germany produced the most selenium that year, producing 1120, 740 and 300 metric tons, respectively [2].

Se belongs to a group of elements that cannot be classified distinctly as either metals or non-metals. It is found in the group VIA as a partner to sulfur (S). Se types are determined by the potential of hydrogen (pH) and measurement of electrical potential [3,4]. It exists in nature in four oxidation states: elemental selenium (Se(0)), selenide (Se(II)), selenite (Se(IV)), and selenate (Se(VI)) [5–7]. According to previous studies [8,9], Marco Polo initially described Se poisoning in the 13th century. However, it was not until research by Schwarz and Foltz that Se's essential function in preventing liver damage in rats was recognized [10].

In the human body, Se plays a vital function as a component of enzymes [11]. Selenium's relevance is attributed to its presence in selenoproteins [12,13]. The significance of Se is also demonstrated in its ability to change the expression and activity of over 25 selenoproteins involved in oxidative stress, detoxification, transport processes, metabolism, and inflammatory responses [14,15]. Although essential, Se is termed as a "two-edged

sword" [16] because of its ability to be both beneficial and detrimental at different concentrations [17]. Se-deficient diets have been linked to various health problems [18], and they are common in many parts of the world [19]. At least one billion individuals globally are Se deficient [20,21]. For instance, the consumption of Se in China was found to be approximately 26.63 µg/d, which is very low [22]. Due to the fact that Se is a non-renewable resource, this worrying situation is expected to get worse in the future.

To increase the Se status in humans and reduce the effects of Se deficiency, the production of crops and animals with increased Se levels is vital. Selenium's necessity for plants is still up for contention [23,24]. Nonetheless, several studies have shown that Se is beneficial for plants and animals [8,25–32]. The biofortification of plants and animals with Se has been researched using multiple techniques including genetic biofortification and the application of selenium fertilizers. Although some are already well known, others are constrained by government regulations and other factors. This review's goal is to provide an overview of the state of Se biofortification research, methods, effects, and challenges. The review focuses on (i) Se biofortification of plants and animals; (ii) Se biofortification effects on plants and animals; (iii) Se biofortification strategies; (iv) Se biofortification and human health benefits; and (v) Se biofortification challenges.

## 2. Methodology

In this study, a comprehensive search was conducted on the World Wide Web for published, peer-reviewed research and review articles utilizing a variety of databases and search engines, including but not limited to Google Scholar, Web of Science, PubMed, Science Direct, Scopus, Directory of Open Access Journals, and MEDLINE. These databases are well-known collections of peer-reviewed articles and widely used. Keywords, index terms, and combinations thereof, such as selenium, selenium biofortification strategies, selenium biofortification, selenium in plants and animals, effects of selenium on plants and animals, selenium and human health, dietary selenium, selenium in food, selenium and cancers, selenium overdose, and selenium benefits, were utilized. Over 1000 studies were discovered using the specified keywords and index terms. The list of studies was then scrutinized and duplicates were removed using endnote, leaving 316 articles. To further uncover related studies, relevant papers cited in the selected publications were reviewed.

## 3. Sources and Pathways of Se

Se sources can be anthropogenic, geogenic, or both [33]. Gypsum, marlstone, volcanic eruptions, sea spray, the weathering of Se-rich rocks, soils, and animal transport are some of the natural sources of Se [34–36]. Atmospheric discharge is one of the most significant sources of Se in different types of soils, as natural resources volatilize Se into the atmosphere [37,38]. In clay soils, Se levels range from 0.8 to 2 mg/kg, whereas tropical soils have a range of 2 to 4.5 mg/kg [39]. In diverse soils, however, Se levels vary from 0.01 to 2 mg/kg [16]. Some studies show that Se accumulates more readily in igneous rocks than in other rock types [35–40].

Se from sediments is transported into rivers and other water bodies by fluctuations in water flow or benthic agitation (Figure 1). In certain areas, Se levels in water from wells and subsurface waters used by humans and livestock for drinking and other activities may surpass 10–20 µg/L, with some concentrations reaching hundreds of micrograms per liter [41]. These waters are not often thought of as an excellent source of Se [42]. Nonetheless, their use results in the transfer and transport of the element in the environment. Farming and industrial activities are the main anthropogenic sources and pathways of Se [43]. However, only around 5% of the overall demand for Se is used by agriculture [44], where it is used in producing fertilizers and animal feeds, among other uses. This renders industrial use the primary anthropogenic source and pathway of Se.

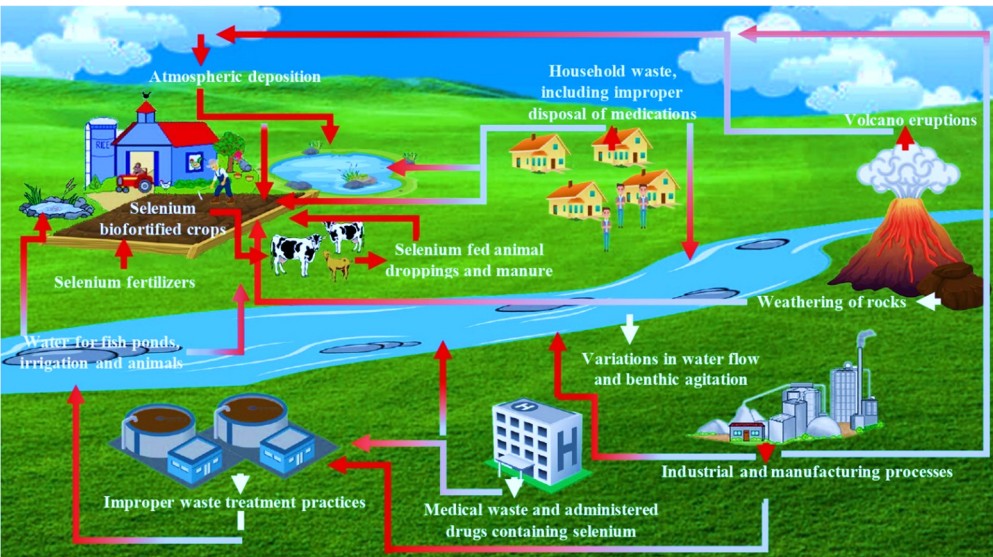

**Figure 1.** Sources and pathways of selenium in the agro-environment.

## 4. Se Biofortification Strategies

Biofortification is a quick, efficient, and sustainable way to lower micronutrient deficiencies [45]. The strategies of biofortification are the approaches employed to achieve biofortification (Figure 2a). Plants and animal produce are classified as biofortified when there is an increase in the Se content in the edible portions of plants and food animals.

### 4.1. Se Forms

As noted earlier, Se exists in soils in four different oxidation states that determine its behavior, specifically its mobility and bioavailability in the natural environment. Some common organic Se compounds include selenomethionine (SeMet), selenocysteine (SeCys), dimethylselenide, selenium methylselenocysteine, dimethyldiselenide, dimethylselenone, methane selenol, and dimethylselenyl sulfide [46]. Se(IV) and Se(VI) are the dominant forms of Se and are often considered the only abundant forms for plant uptake in many studies [47,48]. Se(VI) is water soluble, while Se(IV) is less water soluble and more attached to soil minerals and organic matter [47]. Metallic Se(II) and Se(0) are generally not water-soluble [48]. Se(IV) and Se(VI) are considered the most bioavailable forms for plant uptake due to their solubility [49]. Se(VI) has a higher rate of translocation from roots to shoots compared with Se(IV) [50]. This is because Se(IV) is quickly converted into organic forms like SeCys or SeMet in roots [51]. In anaerobic soils, Se(0) and organic Se(II) are the dominant forms, while Se(IV) and Se(VI) are common in aerobic soils [52]. Se(0) and metallic Se(II) are not water-soluble and, therefore, not bioavailable for plant uptake [52]. Under low redox potential conditions, Se(IV) and Se(VI) can be reduced to Se(II) and Se(0) [53]. Se(0) can also be oxidized into bioavailable inorganic Se compounds through microbial oxidation and hydrolysis [54]. The uptake and transport of Se by plants varies among species and genotypes. The mobility of Se in wheat and canola plants is in the following order: selenate > SeMet > selenite/SeCys [55]. Studies show that rice grains contain higher amounts of Se compared with maize and wheat grains [56]. This may be due to the existence of high-Se and low-Se varieties of rice [57]. The use of Nano-Se to increase the Se content in food is considered a potential solution due to its high biological activity, bioavailability, low toxicity, and large surface area [58]. The use of Se nanoparticles is a promising alternative to other forms of Se as it simplifies application and leads to improved antioxidant metabolism, agronomic sustainability, and waste reduction [59].

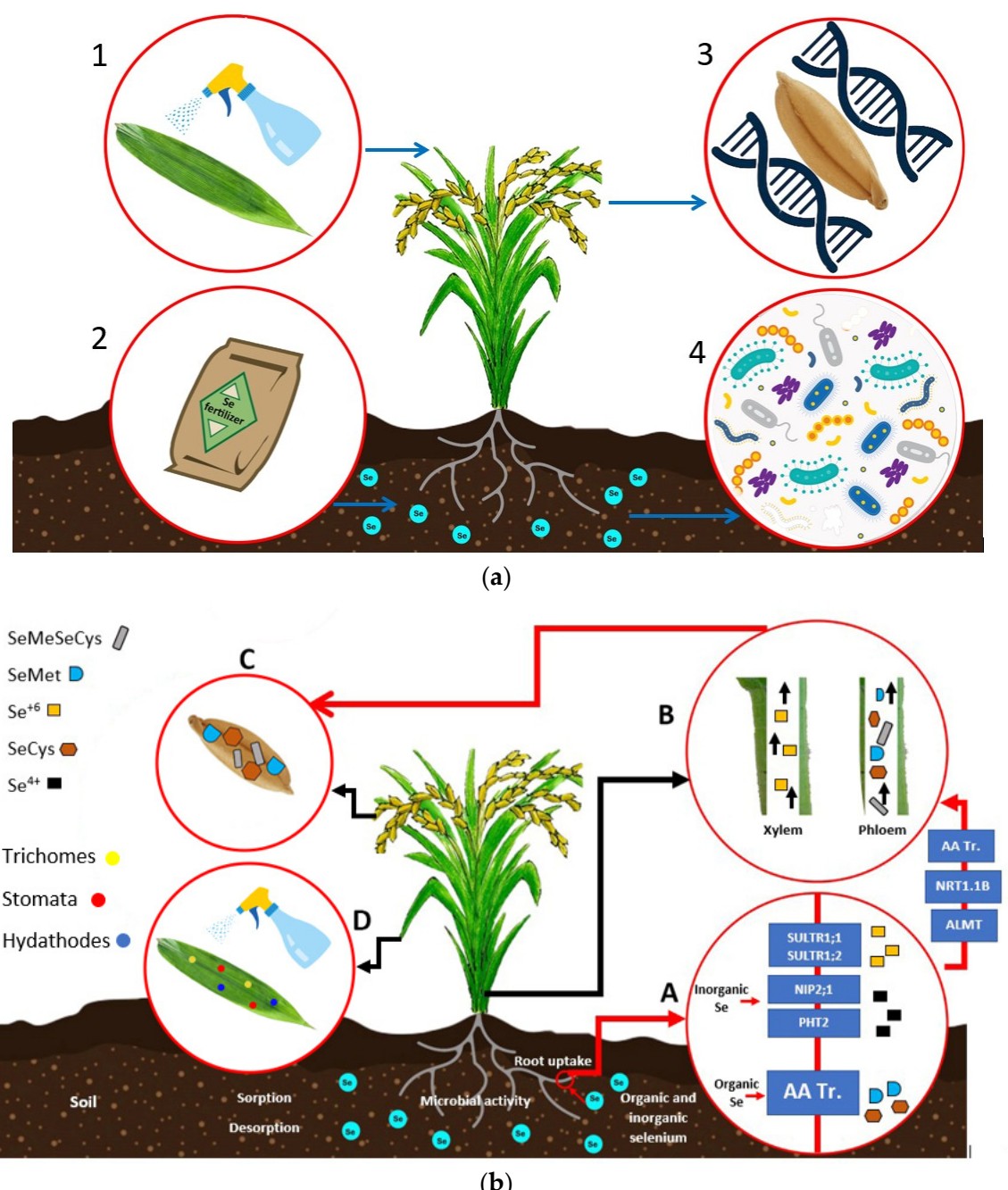

**Figure 2.** (**a**) Selenium biofortification strategies. Note: 1. Foliar application; 2. Soil application; 3. Genetic biofortification; 4. Microbial-assisted biofortification. (**b**) Selenium uptake and transport. (**A**) Amino acid permeases (AA Tr.) and SULTR1;1, SULTR1;2, NIP2;1, PT2, and PT8 transporters aid organic and inorganic Se absorption by the roots. (**B**) The shoots receive organic Se forms from the AA Tr. transporter. (**C**) Selenate is transported by both the xylem and phloem; concomitantly, organic Se compounds enter the seed via the phloem. (**D**) Upon foliar application, Se enters the plants via trichomes, stomata, and hydathodes.

In livestock production, Se is added to animal feed in both organic and inorganic forms [60]. Ruminants absorb and retain organic forms of Se more effectively than inorganic forms. A common way to enhance animal diets with Se is through in-feed administration of Se-enriched yeast, which has a moderate to high Se content and is a source of SeMet [61]. A safe and natural way to provide animals with Se is by offering feed with optimal Se content, as long as the level of Se in the dry matter is carefully monitored. Plants accumulate Se

primarily in the inorganic form and then synthesize seleno-amino acids in SeMet, becoming a source of organic Se for animals [62].

*4.2. Se Biofortification Strategies in Plants*

4.2.1. Foliar Application

Foliar application appears to be the most popular method of applying selenium among all methods because of its simplicity and preferable outcomes. The danger of environmental contamination also seems to be lower. Studies have demonstrated that foliar spray entails minimal use of Se salts [63,64]. This technique entails spraying a crop's leaf surface with a Se-containing solution. Selenium enters the plant through the leaf cuticles. Particles can also enter plants through trichomes, stomata, stigma, and hydathodes [65]. In this respect, soil chemistry and microbiological processes have less of an impact on Se, resulting in a higher absorption rate with modest quantities of administered Se solution. With this strategy, there are changes in plant-specific parameters that must be taken into account, including the quantity of Se applied, leaf area and surface structure, and leaf structure. Wang et al. [66] recently found that applying Se as Se(IV) or Se(VI) using foliar spray during the prefilling stage has a substantial influence on Se concentration in wheat grains. In a previous rice study, foliar application of Se (30–300 µg Se/ha as $SeO_3^{2-}$ or $SeO_4^{2-}$) raised the Se concentration and other bioactive molecules in rice grains [67]. Per Lidon's recent report, foliar $SeO_3^{2-}$ fertilization caused a 427–884-fold increase in grain Se content in four rice genotypes, while $SeO_4^{2-}$ application led to a 128–347-fold increase in grain Se concentrations [68]. Pannico et al. [69] found that foliar spraying of Se (0–40 µM) enhanced leaf Se content in two lettuce cultivars with varying pigmentation, with the red cultivar storing 57% more Se than the green cultivar. However, all treatments decreased the fresh weight of green lettuce by 9%, whereas 32 and 40 µM lowered the fresh weight of red lettuce by 11% and 22%, respectively.

4.2.2. Soil Application

This technique involves amending the soil with Se to raise the amount of overall or bioaccessible Se, enhance the rhizosphere conditions for soil crops, and raise the Se content of produce. With this approach, Se is applied either as Se salts, Se solution, or Se-containing fertilizers. Soil chemistry and microbial activities affect whether Se administered with this technique will result in a desired effect. This strategy is said to have been employed by the Finnish government to boost the population's daily consumption of selenium [70,71]. Soil Se application has been shown to have a favorable influence on various plant physiological systems. Plants absorb Se in the form of organic Se (SeCys and SeMet), Se(IV), and Se(VI) [72,73]. Although plant roots cannot absorb Se(II), they may do so for organic Se species such as SeCys and SeMet and inorganic Se species such as Se(0), Se(IV), and Se(VI) [74,75]. Se (VI) has been shown to enter plants through the sulfate transporters SULTR1; 2 and SULTR1 [72], while Se(IV) enters plants via phosphate transporter transport [76,77]. OsPT2, a phosphate transporter, has been demonstrated to be involved in plant uptake of Se(IV) [78]. Se in the form of Se(VI) applied via soil was the best strategy for increasing Se content in the radish without causing damage to biomass growth. The researchers found that the accumulation of Se in the leaf, root, and whole plants was higher when Se was applied via soil compared with the foliar application [79]. This result is consistent with the findings of a recent study with the same Se application strategy, which found that Se considerably enhanced the Se concentration in mushroom fruit bodies ($p < 0.05$) [80]. However, there was no significant increase in fruit production. Consistent with that study is one that reported a rise in Se concentrations with the same Se application strategy but also no increase in yield [81].

4.2.3. Microbial-Assisted Biofortification

Agronomic biofortification strategies are not always successful due to a number of factors including impromptu rainfall in the case of foliar applications and pH, and heavy

metals in soil applications. Plant growth characteristics and yield have been documented to be influenced by microorganisms located in the rhizosphere via a range of processes. These processes include the release of hormones, nutrient transformations, and stress mitigation [82,83]. The roles played by microorganisms in this respect may be species and/or Se species and bioavailability dependent. Bacterial species such as *Bacillus*, *Entrobacter*, *Paenibacillus*, and *Pseudomonas* have been shown to be capable of Se transformations via methylation and oxidation reduction processes [52,84]. Previous studies found that inoculating wheat with Se-tolerant bacteria derived from Se-deficient soils increased tissue Se accumulation [85]. Researchers demonstrated in 2015 that various bacterial consortia increased Se concentrations in Indian mustard growing in seleniferous soil [86]. A study in 2019 indicated that in a test of two Se forms (SeCys and $SeO_4^{2-}$) in shallots with and without inoculation of arbuscular mycorrhizal fungi, inoculation increased the concentration of Se in the bulb by more than five times [87]. Recently, *Enterobacter* sp. EG16 (7.65107 CFU/mL) was observed to promote the growth and development of pak choi. Chlorophyll concentration, SOD, CAT, and POD activity were likewise increased by the same Se and EG16 doses [88]. These investigations demonstrate the essential roles microbes play in Se biofortification. The study of bioreduction of selenate or selenite using microorganisms such as bacteria, fungi, and plant extracts has become a popular area of interest for scientists [89]. Microbes have been shown to produce the purest form of Se, with previous research demonstrating the production of Se(0) by the anaerobic bacterium *Bacillus selenireducens* [90]. Yeast and other microbes play a vital role in synthesizing Se-containing compounds such as SeCys and SeMet [91]. According to several authors' studies on SeMet determination, it may make up to 90% of the total Se content in yeast cells [92,93].

### 4.2.4. Genetic Biofortification

Genetic engineering has the capability to enhance the capacity of plants to accumulate selenium as an alternative to agronomic approaches. Nevertheless, due to the stringent limitations on the usage of transgenics that are still present in several nations, genetic engineering is still not as prevalent and recognized as agronomic biofortification [94]. However, various chromosomal loci linked to elevated Se accumulation in a number of crops have been reported [74,95,96]. Marker-assisted breeding can be employed to transfer high-Se chromosomal loci from high-yielding, low-Se edible plant varieties into the breeding population [75]. According to Schiavon et al., a significant drawback of conventional or marker-assisted plant breeding is that it must be supplemented with agronomic biofortification treatments employing Se fertilizers when crops are grown in low Se regions [97]. The majority of plants targeted by this method are staple crops [98]. According to a previous study, double-transgenic crops produced from crossed-transgenic mustard greens absorbed up to nine times more selenium than wild-type plants [99]. By using the low Cd replacement line $CSSL^{GCC7}$ as the breeding material, researchers at the China National Rice Research Institute recently reported that $CSSL^{GCC7+GSC5}$ demonstrated increased Se concentrations in grains when crossed with CSSLs containing other major quantitative trait loci for essential mineral elements [100].

### 4.2.5. Crop Breeding

Some researchers believe that conventional crop breeding may be a sustainable and long-term approach to crop biofortification with Se [101]. However, compared to genetic biofortification crop breeding is a slower and less accurate method as this procedure is generally executed by hand. For instance, one study lasted for about five years [102]. Additionally, establishing appropriate and viable genotypic variation may be difficult [103]. Nevertheless, it can be utilized to create new plant types with enhanced features. Crop breeding for Se biofortification uses the conventional procedure of cross-pollinating two separate plants to develop a new hybrid plant with a mix of features from both parents to promote Se absorption and translocation to edible portions of the crop. The researchers' goal in the above-mentioned study was to breed Se-rich red glutinous rice and evaluate the

concentration of Se and protein in various parts of the rice. The red glutinous rice attained an Se concentration of 121.75 ng/g ($\pm$3.01 ng/g) after five years of breeding. According to the results, upwards of 80% of the Se in the grain was organic Se, and over half of the total Se was in the endosperm [102]. The study suggests that plant breeding for Se biofortification was successful and supplementary research is needed in the future.

*4.3. Se Biofortification Strategies in Livestock*

In livestock, biofortification comprises employing agronomic or biotechnological techniques to increase the quantity of vital nutrients in edible sections of animals [104]. Se fertilization of farmlands, dietary supplementation via feed concentrate rations, and direct administration, including injections, are viable supplementation techniques. High Se feed concentrations are expected to raise Se concentrations in livestock. Animals can generate a variety of selenoproteins, including glutathione peroxidase, selenoprotein P, selenoprotein W, thioredoxin reductase, and other iodothyronine deiodinases, using absorbed selenium forms [105]. However, excessive Se ingestion in livestock (5–50 mg per kg of mass) may cause alkali disease, characterized by hoof deformities, a lack of vitality, anemia, and stiffness [106].

## 5. Se Biofortification in Plants

Even though livestock products contain significantly higher amounts of Se, crops are relatively good sources of Se due to their greater bioavailability [107]. While organically cultivated foods are regarded as safe and wholesome, their low Se content may have an unfavorable effect on their appearance [108]. Se supplementation of diets is essential to control Se deficiency [109]. Se fertilization should be performed once every planting season since plant supply can only be maintained for one growing season during crop production for biofortification [110,111]. Sorption, desorption, precipitation, dissolution, production of inorganic and organic complexes, methylation to volatile Se compounds [112,113], and microbiological activity [114] all influence Se mobility and availability to plants (Figure 2b). The amount of Se in a plant is determined mainly by the plant species; soil type in which the plant is produced; use of herbicides, manure, and fertilizers; and agro-ecological management practices [93,115]. Some researchers found that vegetables contain approximately 6 mg/g Se when grown in seleniferous soil; however, asparagus and onions can accumulate up to 17 mg/g Se when grown in similar soils [116].

Se is a valuable element for plants because it stimulates plant development (Table 1) [23,31,32,117]. A recent study [118] discovered that Se improved the plants' agronomic parameters when applied alone. Se application techniques influence whether the element impacts plant development and biofortification. Selecting the appropriate chemical form ($Na_2SeO_4$ or $Na_2SeO_3$) [119] and application approach is a crucial step for achieving desirable effects and efficient biofortification.

*5.1. Se Biofortification Effects in Plants*

Plants have evolved a system in response to oxidative stress. It is an enzymatic antioxidant complex that uses antioxidant enzymes such as superoxide dismutase (SOD), peroxidase (POD), and catalase (CAT) to control oxidative stress [120]. For instance, malondialdehyde (MDA), an oxidized output of membrane lipids, is an indicator of the degree of oxidative stress and lipid peroxidation. Se was shown to modulate antioxidant enzyme activity [120,121]. The primary mechanisms used by Se include the prevention of plant peroxidation, restoration of cell membrane integrity and function, modification of antioxidant enzyme activity, and repair and rebuilding of chloroplast [122].

5.1.1. Se and Salinity

When plants are grown in soils with high levels of salt, their development and growth are stifled. However, the specific mechanism of Se-mediated salinity tolerance is not fully understood. Nonetheless, scientists frequently employ the usage of mineral elements to

enhance crop tolerance to salt-induced stress [122]. Selenium efficacy in preventing such stress has been documented in a number of publications. Onions grown on silt loam soil with a salinity of 8 dS/m were less affected by salt stress after receiving an application of Se in the form of sodium selenite (0.5–1 kg/ha). Improvements in qualitative and physiological markers were also noted by the authors [123]. Se was shown to reduce the negative effects of salt stress in a separate experiment using sunflower plants treated with $Na_2SeO_4$ (5 mg/kg). Treatment with Se resulted in increased glutathione peroxidase activity and a decrease in MDA levels in plant tissues [124]. Numerous studies have shown similar results, indicating that the MDA content in various plant species under different circumstances might be reduced by Se [125–127]. Recently, the use of Se was found to boost antioxidant enzyme activities in grapes, leading to a reduction in salt stress [128]. According to their research, foliar applications of Se treatment ($Na_2OSe_4$: 5–10 mg/L) boosted CAT activity in both groups of plants (0 or 75 mM NaCl), and leaf ascorbate peroxidase activity also increased.

### 5.1.2. Se and Heavy Metals

Plants are capable of absorbing nutrients from the soil for use in their vital metabolic activities. Meanwhile, because there is no particular preference for this mechanism, plants face the possibility of absorbing substances that are detrimental to their physiology. Cadmium (Cd) is one of the most hazardous heavy metals. Previous studies have shown that it has no identified benefits to the environment, and its introduction even in minute concentrations may have negative consequences [129–131]. One recent study showed that POD increased by 28%, 32%, and 27% in the presence of 0.0125, 0.025, and 0.05 mM foliar nano-Se, whereas MDA fell by 8%, 18%, and 4%, respectively. The authors claim that this improved rice's ability to endure Cd stress [132]. Another recent finding suggested that Se enhanced the antioxidant system in tall fescue plants to reduce the negative impacts of Cd and enhance Cd resistance. According to the researchers, Cd treatment (30 mg/L, as $CdSO_4 \cdot 8/3\ H_2O$) increased MDA content by 63% and the relative electrolyte leakage value by approximately 166% higher than that of the control in tall fescue plants. However, Se supplementation (0.1 mg/L, as $Na_2SeO_3$) reduced the MDA content by 52% and the relative EL value by approximately 29%. Additionally, Se treatment considerably increased the CAT activity by 40% and SOD activity by 30% compared with the Cd stress and control, respectively [133]. A study on the interactive effects of Cd and Se on the growth of rice plants found that at a constant Cd content of 4.16 mg/kg, dry grain weight rose considerably with increasing soil Se concentration [134].

In a different study, the researchers found that the root As content decreased by 7 to 55% in the Se treatments (Se-yeast and Se-malt); additionally, the stem As content decreased by 35 to 50%, and the leaf As content decreased by 0.1 to 33%. When Se malt was used, the results showed a similar pattern [135]. The result of Se and heavy metal interaction is thought to rely on Se speciation, relative dosage, application time, and application manner [136,137].

### 5.1.3. Se and Extreme Temperatures

Despite extensive research into how plants react to extreme temperatures, the mechanism underlying the tolerance capacity brought on by Se biofortification remains not fully understood. Under extremely cold conditions, Se (foliar, $Na_2SeO_3$, 5 mg/L) decreased the net photosynthetic rate and chlorophyll content and increased the MDA and hydrogen peroxide contents of strawberry seedling leaves. Se also increased the activities of catalase and superoxide dismutase [138].

Hasanuzzaman et al. looked at the protective function of Se (25 μM $Na_2SeO_4$) in reducing the harm that high temperature (38 °C) caused to rapeseed. They claim that heat-treated seedlings supplemented with Se experienced a considerable reduction in lipid peroxidation as well as an increase in chlorophyll content and antioxidant activity [139]. The development and physiological tolerance of lamb's lettuce cultivated under heat stress (35/22 °C; day/night) by biofortification with Se (foliar (50 mg Se/dm$^3$) and soil ($Na_2SeO_4$)

improved plant growth, reduced oxidative stress due to increased guaiacol peroxidase and catalase activity, and increased levels of GSH, and showed no change in the concentration of phenolic compounds [140].

### 5.1.4. Se and Photosynthesis

The impact of Se on plant leaf anatomy has been explored, with results varying based on soil Se concentration, plant species, and growth stage. Se can impact various physio-biochemical processes, increasing photosynthesis efficiency (Fv/Fm) in chlorophylls and activating the antioxidant system, and improving photosynthesis in stressed plants [141,142]. Wheat Fv/Fm was significantly reduced under salinity stress, but Se and Se + Si application reduced the negative effects on photosynthesis by decreasing the production of ROS that inhibit photosynthetic pigments [142]. In a study on Se's impact on tomato plant photosynthesis and ultrastructural changes under cadmium stress, Se application improved photosynthetic attributes such as leaf transpiration and $CO_2$ assimilation rate compared to the control plant [141]. Se application was also found to increase photosynthetic attributes in sorghum, likely due to its ability to reduce ROS production, repair damaged chloroplasts, and stimulate the production of other vital metabolites [120,143]. A recent study also revealed that both forms of Se (1 μM $Na_2SeO_3$ and $Na_2SeO_4$) led to an increase in mesophyll intercellular spaces and thicker leaves [141].

**Table 1.** Se biofortification effects in plants.

| Plant | Se Forms and Dosage | Se Effects | Reference |
|---|---|---|---|
| Maize | $Na_2SeO_4$ <br> 20 and 40 mg/L <br> Foliar Spray | Increased plant development due to higher salt tolerance during the reproductive stage by reducing oxidative damage and enhancing the activity of antioxidant enzymes. | [122] |
| | $Na_2SeO_4$ <br> 40 mg/L <br> Foliar Spray | Increased fodder yield by 15% | [144] |
| | $Na_2SeO_4$ <br> 0.8–1.0 g/L <br> Foliar Spray | At the jointing stage, fresh ear yield went up by 2.3%; at the large bell stage, it went up by 2%. | [145] |
| | $Na_2SeO_3$ <br> 1, 5 and 25 μM <br> Addition to nutrient solution | Enhanced salt resistance via changes in photosynthetic capacity, antioxidant activity, and $Na^+$ homeostasis | [146] |
| | $Na_2SeO_3$ <br> 5–15 μM <br> Addition to nutrient solution. | Improved the activity of antioxidant system components | [147] |
| Wheat | $Na_2SeO_4$ <br> 0.4 mg $Na_2SeO_4$/kg soil <br> Direct soil application | Height and weight of the plant increased | [147] |
| | $Na_2SeO_4$ <br> 5 μM <br> Direct addition to soil | In normal and NaCl-stressed seedlings, Se increased proline and sugar build-up and supplied additional osmolarity to preserve relative water content and safeguard photosynthesis. | [148] |
| | $Na_2SeO_4$ <br> 10 mL/pot <br> Foliar spray | Enhanced antioxidant enzyme activity; improved plant growth, photosynthetic capacity, relative water content, and chlorophyll content | [149] |
| Rice | $Na_2SeO_3$ <br> 25μM <br> Addition to nutrient solution. | Increased phenolic chemicals and decreased arsenic accumulation | [150] |
| | $Na_2SeO_4$ <br> 10 μM <br> Addition to nutrient solution. | Increased plant growth and biomass, and increased protein content. The activities of MDA, $H_2O_2$, APX, CAT, and SOD reduced in the shoots. | [151] |
| | $Na_2SeO_3$ <br> 0.8 and 1.0 mg/L | As-induced toxicity significantly decreased germination by 70%, and Se supplementation by seed priming increased germination by 9% and root, shoot, and seedling biomass accumulation by 1.3, 1.6, and 1.4 folds, respectively. | [152] |

**Table 1.** *Cont.*

| Plant | Se Forms and Dosage | Se Effects | Reference |
|---|---|---|---|
| Tomato | Na$_2$SeO$_3$ or Na$_2$SeO$_4$ <br> 1 μM <br> Addition to nutrient solution | Enhanced photosynthesis and increased root and shoot dry weight | [141] |
| | Na$_2$SeO$_3$·5H$_2$O <br> 10 μM <br> Direct soil application | Increased the levels of stomatal conductance, chlorophyll and carotene, transpiration rate and net photosynthesis rate | [125] |
| | Se nanoparticles <br> 10 mg/L <br> Foliar Spray | Increased the yield by 21% | [153] |
| Pepper | Na$_2$SeO$_3$ <br> 5 μM <br> Addition to nutrient solution | Increased root development, membrane stability index, chlorophyll concentration, and starch content in leaves | [154] |
| | Na$_2$SeO$_3$ <br> 3 and 7 μM <br> Direct addition to soil | Plants cultivated in the medium containing 0.25 mM Cd had higher mean productivity, a greater capacity to withstand stress, and a higher yield stability index when the Se doses were added. | [155] |
| Onion | Na$_2$SeO$_3$ <br> 0.5 and 1 kg/ha <br> Foliar spraying | Improvements in both qualitative and physiological markers. Maximum production at 1 kg/ha of foliar Se supplementation | [123] |
| Garlic | Na$_2$SeO$_4$ <br> 4, 8 and 16 mg/L <br> Addition to nutrient solution. | Se improved salt tolerance and decreased oxidative damage by boosting the activity of antioxidant enzymes. | [156] |
| Cucumber | Na$_2$SeO$_3$ <br> 2 g/L <br> Addition to nutrient solution | Increased root and shoot biomass, as well as chlorophyll content | [157] |
| Mustard greens | Na$_2$SeO$_4$ <br> 4 μM/kg <br> Addition to nutrient solution | Improved growth, increased chlorophyll and carotene content, net photosynthesis rate, stomatal conductance, and transpiration rate | [158] |
| Broad Beans | Na$_2$SeO$_3$ <br> 1.5 μM <br> Addition to nutrient solution | Decreased MDA content; and H$_2$O$_2$ buildup, increased chlorophyll content shoot elongation and shoot fresh weight | [159] |
| Lemon balm | Na$_2$SeO$_3$.5H2O <br> 0.2 μM <br> Addition to nutrient solution | Enhanced growth | [160] |
| Strawberry | Na$_2$SeO$_4$ nanoparticles, 10 and 20 mg/L <br> Foliar Spray | Increased number of fruit plants−1 by 21.22 and 12.54%, and yield by 21 and 14%, respectively, in two growing seasons | [161] |
| Pomegranate | Na$_2$SeO$_4$ and Se-nanoparticles, 1 and 2 μM <br> Foliar Spray | In two growing seasons, the number of fruits per tree grew by 1.35 and 1.28 times, and the yield grew by 1.17 and 1.16 times. | [162] |
| Cowpea | Na$_2$SeO$_4$ <br> 5 and 10 μM <br> Foliar application | Enhanced yield-related indicators, growth, and protein levels | [163] |
| Sunflower | Na$_2$SeO$_4$ <br> 5 mg/kg <br> Direct soil application | Increased antioxidant enzyme activity | [124] |
| Tobacco | Na$_2$SeO$_3$ <br> 0.1 mg/L <br> Addition to nutrient solution. | Se reduced the toxicity of the high As dosage (5 mg/L) and stimulated the development of the plant by increasing antioxidative stress resistance and decreasing MDA levels. | [164] |

## 6. Se Biofortification in Livestock

Rapid livestock production increases the necessity for Se [165,166]. In animals, the liver, heart, and skeletal muscle are the first organs to become deficient in Se [114]. Inadequate levels of Se have been linked to a variety of illnesses in livestock, including nutritional myopathy and ill-thrifting [167,168] as well as white muscle disease [169], which is primarily unnoticeable in older animals [170]. White muscle disease is a severe degenerative condition marked by rigidity, exhaustion, limb trembling, and inflamed muscles [171].

It has been labeled the animal variant of Keshan disease [172]. Ruminants' absorption of Se is less efficient and complicated than that of non-ruminants since rumen microbial populations may convert Se to inaccessible forms [173], and only one-third of inorganic Se is absorbed. Some studies suggest that Se supplementation in animal feeds should be between 0.05 and 0.1 mg/kg dry matter (DM) to satisfy Se requirements [174,175]. Freshwater creatures such as crayfish, crabs, and carp [176] may deposit Se in their tissues, even in rare Se geographic locations. However, it is uncertain whether eating seafood has a role in their Se buildup.

*Se Biofortification Effects in Livestock*

One of the most critical factors recognized in the pathological course of many illnesses and cancers is oxidative stress and the unregulated creation of reactive oxygen species [177]. Total superoxide dismutase (T-SOD), GSH-Px, and catalase are intricate defense and repair mechanisms that may protect animals from oxidative damage [178]. Livestock research has shown that Se may reduce heavy metal toxicity and alter the degree of heavy metal exposure and illness (Table 2). For instance, in a previous study, Se reduced Cd-induced hepatotoxicity [179]. The researchers discovered that in cocks, a 10 mg/kg Se(IV) diet decreased Cd buildup and boosted antioxidant resistance in hepatic tissue, as well as Cd-induced morphological alterations and oxidative stress. According to the researchers, the results can be explained by Se's crucial role in avoiding lipid peroxidation and preserving the structural and functional integrity of tissues. Selenium supplementation protected the chicken brain against chromium damage by blocking adverse effects [180]. They further found that Se supplementation reduced MDA activity and that the supplementation at 5 mg/kg BW greatly boosted SOD activity. Furthermore, Se supplementation reduced MDA activity because Se inhibits hydroxyl radical production and maintains tissue function. Additionally, Se is acknowledged to be a crucial component for reproductive function in poultry. Dietary Se supplementation at varying concentrations (0.10–1.00 mg/kg) and sources enhanced laying performance and egg quality [181]. Se administration using either organic (Se-enriched yeast) or inorganic (sodium selenite) forms of Se at levels of 0.3 and 0.15 mg/kg against the H9N2 virus dramatically reduced viral shedding within the chicken, with the organic form proving more efficient [182]. However, according to an experiment on hens augmented with Se orally at doses of 5, 10, and 15 mg/kg for 15, 30, and 45 days, high intakes of Se caused a significant decrease in the levels of the cytokines IFN-$\gamma$ and IL-2 in both serum and the thymus, as well as a low-to-moderate incidence of pathological changes in the thymus tissue, which suggests a decrease in protection and an upsurge in oxidative damage [183].

A study by Li et al. [176] revealed that nano-Se may counteract the oxidative stress and inflammation in chickens caused by di-(2-ethylhexyl)phthalate (DEHP), a plasticizer extensively used in the food sector. According to their study, the SOD, T-AOC, GSH-PX, and CAT activities of the DEHP group were considerably reduced ($p < 0.05$) compared with those of the control group, whereas the SOD, T-AOC, and CAT activities of the nano Se group were significantly increased ($p < 0.05$).

A previous study in pigs found that 6 mM Se Met suppressed PCV2, a postweaning multisystemic wasting syndrome, and that 2 or 4 mM Se Met prevented the increase in PCV2 replication induced by oxidative stress [184]. Some studies have speculated that the underlying mechanism of Se Met inhibition of PCV2 replication is mediated by increased activity of GSH-Px, which shields the cell from free-radical oxidant harm [185]. The effects of dietary Se on parainfluenza virus infection in lambs have also been studied, focusing on the innate and adaptive immune responses to virus infection. When the parainfluenza virus was injected into lambs that had been fed Se, they exhibited increased immunological activity [186]. A recent study found that health and reproduction parameters improved for cows fed with organic Se forms [187]. Sun and coworkers [188] found that Se-yeast treatment boosts antioxidant capacity in dairy cows. They showed an increase in serum

glutathione peroxidase activity ($p < 0.05$) and total antioxidant capacity ($p = 0.08$), as well as a reduction in MDA content ($p < 0.05$).

**Table 2.** Se biofortification effects in livestock.

| Animal | Se Form and Dosage | Se Effects | Reference |
|---|---|---|---|
| Cow | Se yeast supplement | Enhanced antioxidant levels and immunological responses following calving | [15] |
| | Se-enriched alfalfa hay | Supplemental selenium increased immunization responses against Escherichia coli during the weaning transition phase and subsequent growth and survival in the feedlot. | [189] |
| Pig | DL-selenomethionine 2–16 μmol/L | Significant inhibitive effect on Porcine circovirus type 2 replication | [190] |
| | SeMet 2–6 μM | Inhibited porcine circovirus type 2 replication and its related oxidative stress | [184] |
| | Se yeast diet | Piglets given selenium yeast showed greater digestibility of DM, crude protein, and crude fat; which impacted the production of inflammatory cytokines, and decreased the quantity of Escherichia coli in feces. | [191] |
| Chicken | SeMet | Increased immune function and selenoprotein expression, and reduced the inflammation generated by lipopolysaccharides. | [192] |
| | 0.3 mg/kg Se yeast 0.3 mg/kg of organic Se from *Stenotrophomonas maltophilia* (bacterial organic Se, ADS18). | Bacterial selenoprotein or Se-yeast improved the performance index, egg quality features, egg yolk and tissue of Se concentrations and intestinal villus. | [193] |
| | Se Enriched Yeast $Na_2SeO_3$ (High—0.30 mg/kg of feed; Low—0.15 mg/kg of feed) | Virus shedding from the cloaca was substantially reduced in all selenium-supplemented groups compared with non-supplemented control groups. | [182] |
| | sodium selenite 10 or 20 μg | Se injection enhanced immune and antioxidant responses | [194] |
| | Probiotics as (P, 0.11 mg Se/kg) $Na_2SeO_3$ (SS, 0.41 mg Se/kg) and (SP, 0.41 mg Se/kg) | In groups supplemented with selenium, oocyst shedding and cecal lesion scores were reduced. | [195] |
| Sheep | Se yeast supplementation >4.9 mg Se/week | Supplementation with Se-yeast enhanced the Se status of sheep and the expression of genes involved in innate immunity in whole blood neutrophils. | [196] |
| | Se yeast 0.5–1.0 mg/kg | Drip loss of muscle decreased significantly with an increase in dietary selenium yeast Supplementation. | [197] |
| Rabbit | Se yeast 0.3 mg Se/kg diet | Positive effect on growth performance of rabbits. Se increased daily gain and the final body weight. Supplementation with Se increased muscle Se content to 559% of the control level. | [198] |
| | Sodium selenate solution 10% of Se-fortified olive leaves (2.10 mg/kg) | Meat exhibited better oxidative status and a 5-fold higher Se content compared to that of the other treatments. | [199] |

### 7. Se Biofortification and Humans

Se deficiency in humans can be linked to lower levels in plant and livestock produce. Deficiency in humans can be associated with numerous ailments that denote a significant impact on the socioeconomic development of individuals. However, due to changes in laboratory methodologies throughout the world and the lack of specific biofortification values and guidelines for application and intake, there is a significant variability globally (Table 3). Some regulatory agencies recommend daily Se consumption of 30–85 μg/d for males and 30–70 μg/d for females to meet dietary requirements [200–202].

On average, the concentration of Se in cereals produced in Europe reportedly varies from 0.02 to 0.05 mg/kg DM, whereas in North America, it is 0.2 to 0.5 mg/kg (Table 3) [45,203]. Previous studies [204,205] demonstrated that the highest mean and median amounts of Se in rice grown in the United States were 176 and 180 ng/g, respectively. In contrast, the lowest mean and median concentrations of Se in Egyptian rice were 9.0 and

6.0 ng/g, respectively. The researchers also reported that the average value of rice in India was 152 ng/g; however, 5% of Chinese rice had an Se content of more than 200 ng/g.

Se consumption levels equivalent to the required dietary amount have been reported in some countries. For instance, Belgium and France [41,206] have met the required dietary quota. The present Se intake in Finland is also reported to be in accordance with the Nordic, European Union, and United States standards [207–209]. However, studies from several parts of the world show that the required dietary amount is not met. Places such as Italy and Slovenia showed intakes that were below the recommended dietary limit [210–212]. Previous studies in Kuwait, Saudi Arabia, and Turkey that looked at Se concentration in blood plasma and serum recorded that breast milk and umbilical cord blood were low [213–215]. In Turkey, the Se content in breast milk was below the international standard range (18.5 µg/L) during the breastfeeding period [216]. In Britain, a longitudinal study of British individuals in good health revealed poor Se intake [217]. Se intake in Poles compared to that of Spaniards was reported to be four times lower [218–221].

**Table 3.** Se intake status in different countries.

| Country | Se Intake | References |
|---|---|---|
| Russia | 35.5 µg | [222] |
| Brazil | 84.3–105.9 µg | [223–225] |
| United States of America | 60–220 µg | [209,224,226–228] |
| Turkey | 20–138 µg | [215,229–239] |
| Slovakia | 27–43 µg | [240] |
| Saudi Arabia | 34–121.65 µg | [241,242] |
| Venezuela | 200–350 µg | [21,243] |
| Czech Republic | 10–25 µg | [243] |
| Canada | 98–224 µg | [225,228] |
| England | 12–43 µg | [228] |
| Belgium | 28–61 µg | [224] |
| Germany | 35–47 µg | [224,225] |
| Mexico | 61–73 µg | [224,228] |
| Venezuela | 200–350 µg | [224,228] |
| Australia | 57–87 µg | [209,228] |
| Japan | 104–127 µg | [228] |
| Greece | 110 µg | [228] |
| China | 3–6690 µg | [22,224,243] |
| Poland | 30–40 µg | [244] |
| Finland | 70–80 µg | [71] |
| Spain | 44–50 µg | [117,245] |
| Austria | 48 µg | [21,117] |
| Slovenia | 87 µg | [246] |
| Slovakia | 27–43 µg | [240] |
| Jordan | 59.26 µg | [247] |
| Greenland | 193–5885 µg | [248] |

*7.1. Se in Humans*

7.1.1. Se Intake

Mehdi et al. [116] found that between inorganic and organic Se, the former is more harmful than the latter. Additionally, Vinceti et al. [42] discovered that inorganic Se was 40 times more dangerous than organic Se. However, several studies on humans have revealed that doses of up to 800 µg Se/day provided as Se-yeast did not result in any harmful effects [105]. Several variables influence Se intake, including nutritional habits and geographical location, as well as food imports and sources of food. Reduced Se intake and levels in the United Kingdom and other northern European nations are thought to have originated since the mid-20th century due to changing trade, which resulted in lower wheat imports from the US and Canada [172]. In New Zealand, the process that was used

to remove arsenic from superphosphate fertilizers also removed Se, decreasing the Se status in plants and livestock [44].

Se deficiency is linked to a variety of cancerous diseases [249], renal impairment [250], type 1 diabetes [251], epilepsy, and cardiovascular diseases [93]. Keshan illness [252] was the first human disease associated with Se deficit. Kashin–Beck disease, a bone and cartilage disease described in China, Tibet, North Korea, and Siberia, was the second [249]. Thyroid hormone metabolism problems and selenoprotein N-related myopathy, both of which are congenital muscle disorders, have also been linked to decreased selenoprotein expression [253]. Other variables besides Se insufficiency may be the main cause of the conditions of various diseases and increased oxidative stress. However, a strong Se status along with a sufficient intake of other antioxidative nutrients may assist cells and tissues better withstand the detrimental oxidative stress caused [44]. For example, hyperglycemia [231] or the immune system's response to infection [254].

### 7.1.2. Health Benefits of Se

Se has been shown to aid muscular function by improving endurance and recuperation, as well as slowing the aging process [116]. Cengiz et al. [255] discovered a significant link between low Se levels in expectant women's blood and the occurrence of neural tube abnormalities, particularly anencephaly and rachis. Hence, a reduction in reproduction in females is usually related to Se-insufficient body saturation [256,257]. However, the role of Se in this cycle is unclear.

In 1960, the US states with higher intakes of Se reported lower rates of cancer deaths than states with lower intakes [258,259]. The socioeconomic impact of cancers is significantly increasing. In 2010, the global economic cost of cancer was estimated to be over \$1.2 trillion [238], and it was €199 billion in Europe in 2018 [260]. The ingestion of Se-supplemented foods has been researched as a potential remedy. For instance, the consumption of Se-fortified garlic and broccoli produces Se-methyl–Se-cysteine, which is converted to methyl selenol, a powerful cancer-fighting compound [261,262]. According to a study on 18 individuals given 200 µg Se-enriched broccoli daily for three days, Se intake led to noticeably greater levels of both Th1 and Th2 cytokines released by peripheral blood mononuclear cells. The researchers found that supplementation raised plasma Se levels [263]. The contribution of Se to tumor cell invasion, cell proliferation, and apoptosis has been investigated [264,265]. Se contained in a fraction of selenoproteins, has been found to promote anti-carcinogenic factors and have anti-proliferative and anti-inflammatory properties [266,267]. It has also been shown to lessen the severe side effects of several chemotherapeutic drugs while maintaining their anticancer properties [268,269].

In the treatment of viral infections, 200 µg of selenium per day reduced HIV patients' hospital admissions and infection-related admissions [270]. Similar research showed that higher Se content in serum was associated with lower viral load, even after adjusting for antiretroviral therapy regimen and adherence [271]. However, some researchers have disputed the data analysis approach [272]. In patients who received a live attenuated poliovirus vaccination, treatment using 100 µg Se/d increased the number of total T cells and Th cells and improved virus clearance [273].

Se is reportedly used in a number of health materials. It suppresses bone cancer in a localized location without damaging healthy tissue in the surrounding area, indicating tremendous potential for novel bone cancer therapy options. Depending on the kind of Se used, the frequent technique for treating bone cancer is incorporating it into ceramic substrates [274]. Moreover, Se has been used as an anticancer material in breast, lung [275], and prostate [276] applications. This indicates Se antioxidant capabilities, immunological protection, carcinogen detoxification, cell proliferation modulation, and suppression of cancer cell invasion and angiogenesis, among other things [277]. Selenium has also been shown to lessen the severe side effects of several chemotherapeutic drugs while maintaining their anticancer properties [268,269].

## 8. Se Biofortification Challenges

### 8.1. Influence of Soil Characeristics

Direct soil application has been shown to enhance the biofortification of crops with Se. However, studies highlight issues with Se biofortification through soil, owing to the lower uptake by plants and the probable economic losses associated with this strategy [278]. Previous research, for example, found that improving Se status in grain to 100 µg Se/kg requires almost six times more fertilizer for soil treatment than foliar spray [279]. Plants absorb most of their nutrients in the rhizosphere, and the conditions there can affect how bioavailable Se is to plants [280,281].

Ions present in soils may hinder the uptake of selenium by plants (Figure 3). The anions sulfate and phosphate may compete with Se for absorption by plants [282]. Se remains bound in phosphate precipitates when phosphate fertilizers are applied to the soil, rendering it unavailable for uptake [44]. Selenium also shares similar characteristics to S [283,284], and it is documented that adding S to the soil inhibits plant Se absorption [285,286], as they share the same metabolic pathway during translocation [287].

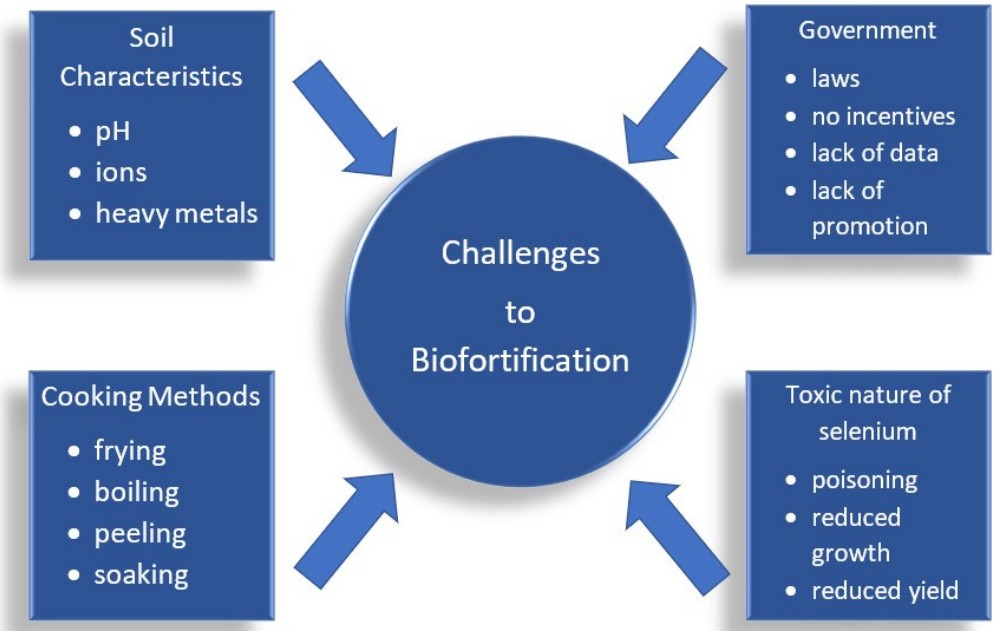

**Figure 3.** Challenges to biofortification.

Additionally, metal and metalloid bioavailability and speciation in soils are affected by soil clay [288,289]. Se is poorly bioavailable in clayey soils as a result of its attraction to clay minerals [290]. According to studies, Se can adsorb on positively charged sites of Al-octahedral sheets in clay minerals such as kaolinite and changes significantly with soil pH [291].

Soil pH also affects the adsorption of Se [282]. Selenite is insoluble in acidic and neutral soils while Se(IV) is more readily absorbed by plants in neutral and alkaline soils [292]. This has been demonstrated in previous studies in which Se-Se(VI)-enriched soils were noted to absorb Se 10 times more readily than Se-Se(IV) [293]. Additionally, according to thermodynamic calculations, Se(IV) should predominate in mineral soils with pH values between acidic and neutral (7.5 < pe + pH < 15), and selenate in alkaline and well-oxidized soils (pe + pH > 15) [294–296].

### 8.2. Food Processing Methods

Consuming Se-fortified foods is critical for improving Se status in humans. The major goal of Se biofortification of plants and animals is to raise the Se status in humans. The concentration of Se in a living organism is heavily influenced by Se consumption [20,297].

However, Se-fortified products such as grains, vegetables, and meat are usually ingested after cooking practices such as roasting, frying, and boiling. Se content in foods is reported to be affected by cooking and preservation methods. Dong et al. [298] found that the overall Se content was reduced by about 43.3% after boiling (10 min, 100 degrees Celsius), 38.5% was lost to the water used, and 31.7% was lost after frying (10 min, 180 degrees Celsius). They also observed that Se in tubers was reduced by 53.4–69.9% when peeled. Some researchers observed that among the five typical processing techniques, frying recorded the highest Se loss (64%) in garlic, and Se volatilization caused by high-temperature heating was the primary factor [299]. Correspondingly, other food processing techniques, such as soaking, have recorded a loss in the concentration of Se content by 2.6% to 7.2% [300]. Boiling, steaming, and frying were found to negatively affect the total Se concentration in wheat (5.6–13.6%) [301]. A recent publication found that the latter had a more significant decreasing effect on the total Se concentration in *Pleurotus eryngii* fruit bodies than in boiled fruit bodies [80]. Additionally, food preservation methods have been observed to cause Se to be embedded in the food medium and not freed in the intestines [302]; as a result, absorption and usage of the element are affected [303]. Matos and coworkers found that while the Se content in blue shark rose after steaming and grilling, its bioaccessibility decreased considerably ($p < 0.05$) [304]. Zhou et al. [304] discovered that 35.3% of Se in boiled tubers was not bioavailable, but the figure for fried samples rose to 76.6% following oral and gastrointestinal digestion.

*8.3. Toxic Nature of Se*

Despite the importance of Se, its toxic nature past a certain level is a factor that hinders its biofortification. The transition between biofortification and toxicity of Se is narrow. Considering the very small range between nutritional quantities that are deadly and inadequate, it is easy for selenium supplementation in animals to lead to toxic or even fatal doses. Treatments for Se biofortification in animals can lead to poisoning as a result of either unintentional or deliberate dosages [305]. In plants, while some can accumulate high concentrations of the element and are termed hyperaccumulators [292], most plants that are consumed, such as wheat, rice, maize and barley, which cannot accumulate such higher amounts and are termed non-accumulators [72]. Selenium toxicity has been reported to cause a negative impact on plant physiology, growth and development [50,306–313], nutrient content [50,312,314,315], and yield [306,316,317].

Selenium ($SeO_3^{2-}$; 50 or 100 μM) produced secondary nitrooxidative stress, lowered root development and yield, lowered cell viability, affected cell wall structure by altering pectin and callose, and lowered stomatal density in a study using thale cress [306]. Se-fortified ($SeO_4^{2-}$ 80 μM and $SeO_3^{2-}$ 20 μM) cucumbers showed reduced biomass, shoot growth, root growth, and leaf area. Additionally, it worsened nutritional content, decreased the formation of photosynthetic pigments, increased lipid peroxidation, and decreased chlorophyll fluorescence [50].

*8.4. Government Support*

Considering that Se deposits are rare, finite, and potentially susceptible to depletion by improper or inefficient use, they must be conserved [279]. Support from government in the type of funds for research and laws will be essential to ensure the proper utilization of the resource. Most importantly, as the greatest advantage of Se enrichment is to improve public wellbeing by lowering illness costs [318], policymakers must ensure that scientists, food producers, and health providers have enough data on the population and environment to aid them in better assessing the situation and providing the necessary and required duties. Education programs about Se biofortification, Se biofortified products and why they are needed will also go a long way to help the social acceptance and patronage of Se-biofortified produce. A higher patronage will likely cause a more affordable price for the general public. According to Bouis et al., [319], if provided biofortified produce is comparatively less expensive than the competition and has similar quality, individuals from developed and

underdeveloped countries will patronize them. However, most developing economies do not have the resources needed to ensure the needed measures for biofortification and put to action.

## 9. Conclusions

Inadequate Se status in plants and livestock appears to be the most prevalent cause of Se deficiency. Se-deficient crops may stem from either a low Se concentration in the soil, a limited availability of soil Se for absorption by plant roots, or both. In food animals, low levels of Se in their diets seem to be the primary cause. Research trends suggest Se supplementation's efficacy in maintaining homeostasis of several metabolic processes. Researchers are utilizing various techniques to increase Se content in edible sections of crops and livestock to address both the issue of low Se intake and its impacts. However, biofortification of agricultural products should be thoroughly researched and assessed in view of Se resource planning and conservation. Additionally, further study is required to confirm the safety and appropriateness of present Se levels to define optimal levels. Modern biological and analytical technologies must also be progressively developed and utilized to assess the imprints and pathways of selenium in a wide variety of foods and geographic locations to find other Se-deficient regions and combat the existing global Se shortfall. Additionally, international support systems must be recognized for developing economies.

**Author Contributions:** Conceptualization, O.P.D., X.Y. and R.Z.; writing—original draft preparation, O.P.D.; writing—review and editing, O.P.D., B.A.-B., Z.Z., J.S., Z.W., X.Y. and R.Z.; visualization, O.P.D. and B.A.-B.; supervision, Z.Z., J.S., Z.W., X.Y. and R.Z.; funding acquisition, X.Y. and R.Z. All authors have read and agreed to the published version of the manuscript.

**Funding:** This research was funded by the Special Fund for Functional Agricultural Development of National Agricultural Parks (No. NJGJNCY-FAST01) and the National Natural Science Foundation of China (No. 41976220).

**Institutional Review Board Statement:** Not applicable.

**Data Availability Statement:** Not applicable.

**Acknowledgments:** We highly acknowledge the support of the Alliance of International Science Organizations (ANSO) and the Nanjing Institute for Functional Agriculture Science and Technology (iFAST). We are thankful to the National Natural Science Foundation of China and the Functional Agricultural Development of National Agricultural Parks.

**Conflicts of Interest:** The authors declare that they have no conflict of interest.

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
