# Peer review of "Selenium Biofortification: Strategies, Progress and Challenges"

_agriculture, doi:10.3390/agriculture13020416_

Round 1

Reviewer 1 Report

The review article entitled (Selenium biofortification: Strategies, Progress and Challenges) Shows the importance of selenium to humans, plants and animals, methods of bio-fortification to each of them, and the advantages of each method. It also deals with the challenges facing selenium bio-fortification.

General comments

The manuscript is well presented and the flow of information is well arranged. It also addresses an important issue closely related to human health and the safety and quality of its food web.

Figures and tables are good and presented properly.

It is better to allocate a section to talk about the comparison between the different forms of selenium, the advantages of using each form and its degree of safety, and adding the nanoscale form Se0 of selenium, which is widely mentioned at the research level nowadays. Of course, some of the aforementioned points were dealt with separately in the review, but allocating a discussion about them in a separate section is preferable.

Minor comments

Section 3.1.3 : There is another role for microbes, and it is very important in providing forms of selenium within compounds such as seleno-cysteine and seleno-methionine, as is the case in yeast. These forms of selenium are the safest and preferred in the case of human or veterinary use.

Line 126: delete repeated brackets   (SeMet))

Line 280: Se (5 mg/L)   Please mention the other conditions related to the process of adding selenium, such as the method of application and the amount of solution used for area unit.

Line 290: delete repeated brackets  (Na2SeO4)).

Author Response

Please, thank you for taking the time to review our article. Please kindly find attached the line numbers addressing the corrections made.

It is better to allocate a section to talk about the comparison between the different forms of selenium, the advantages of using each form and its degree of safety, and adding the nanoscale form Se0 of selenium, which is widely mentioned at the research level nowadays. Of course, some of the aforementioned points were dealt with separately in the review, but allocating a discussion about them in a separate section is preferable.

Please a section about the forms of selenium has been added (Line 96-129, Section 3.1 Selenium forms)

Section 3.1.3: There is another role for microbes, and it is very important in providing forms of selenium within compounds such as seleno-cysteine and seleno-methionine, as is the case in yeast. These forms of selenium are the safest and preferred in the case of human or veterinary use.

Please a section about the role for microbes has been added (Line 196-203)

Line 126: delete repeated brackets (SeMet))

Please the repeated brackets have been deleted (Line 163)

Line 280: Se (5 mg/L)   Please mention the other conditions related to the process of adding selenium, such as the method of application and the amount of solution used for area unit.

Please the correction has been made (Line 339)

Line 290: delete repeated brackets (Na2SeO4)).

Please the repeated brackets have been deleted (Line 349)

Author Response

Please, thank you for taking the time to review our article. Please kindly find attached the line numbers addressing the corrections made.

43-44: The authors reported Se's essential function in preventing liver damage in rats? Write the author.

Please the sentence has been restructured (Line 43 and 44)

59-60: The biofortification of plants and animals with Se has been researched using multiple techniques. Write the name of these multiple techniques here?

Please the names of the techniques have been added (Line 61)

203: Is there any research publication available on Se rich genotypes within the cultivated or their wild relatives of plants? if so, please write a paragraph about “Se biofortification in plants using breeding of Se rich genotypes”

Please a section about crop breeding for Se biofortification has been added (Line 224, Section 3.2.5 Crop breeding)

221: Selecting the appropriate chemical form [97]. Which form?

Please the forms of selenium have been added (Line 281)

486: However, Se-fortified products are usually ingested after cooking. What is it? Please describe

Please the sentence has been restructured (Line 560 and 561)

514-515: Selenium toxicity has been reported to cause a negative impact on plant growth. Is this negative impact causes any loss of nutrients or shelf life of fruits/vegetables or physiology of plant?

Please the sentence has been restructured and references have been added (Line 590 and 591)

Reviewer 3 Report

The authors didn't mention any effect of selenium on the anatomy of the leaf and stem, although there are articles mentioning that.

Author Response

Please, thank you for taking the time to review our article. Please kindly find attached the line numbers addressing the corrections made.

The authors didn't mention any effect of selenium on the anatomy of the leaf and stem, although there are articles mentioning that.

Please the effect of selenium in the leaf has been added under section 4.1.4 Se and Photosynthesis (Line 352, Se and photosynthesis)